# How Well Can Quantum Embedding Method Predict the Reaction Profiles for Hydrogenation of Small Li Clusters?

**DOI:** 10.3390/nano14151267

**Published:** 2024-07-29

**Authors:** Dominic Alfonso, Benjamin Avramidis, Hari P. Paudel, Yuhua Duan

**Affiliations:** 1National Energy Technology Laboratory, U. S. Department of Energy, Pittsburgh, PA 15236, USA; 2NETL Support Contractor, 626 Cochran Mill Road, Pittsburgh, PA 15236, USA; bavrimidis@pitt.edu (B.A.); hari.paudel@netl.doe.gov (H.P.P.); 3Department of Chemistry, University of Pittsburgh, Pittsburgh, PA 15260, USA

**Keywords:** quantum computing, quantum simulator, active space embedding methods, full configuration interaction, coupled cluster methods, hydrogenation reactions

## Abstract

Quantum computing leverages the principles of quantum mechanics in novel ways to tackle complex chemistry problems that cannot be accurately addressed using traditional quantum chemistry methods. However, the high computational cost and available number of physical qubits with high fidelity limit its application to small chemical systems. This work employed a quantum-classical framework which features a quantum active space-embedding approach to perform simulations of chemical reactions that require up to 14 qubits. This framework was applied to prototypical example metal hydrogenation reactions: the coupling between hydrogen and Li_2_, Li_3_, and Li_4_ clusters. Particular attention was paid to the computation of barriers and reaction energies. The predicted reaction profiles compare well with advanced classical quantum chemistry methods, demonstrating the potential of the quantum embedding algorithm to map out reaction profiles of realistic gas-phase chemical reactions to ascertain qualitative energetic trends. Additionally, the predicted potential energy curves provide a benchmark to compare against both current and future quantum embedding approaches.

## 1. Introduction

Theoretical chemistry is a key driver in the development of algorithms for quantum computers which have the potential to solve the expectation value of the electronic Hamiltonian accurately and efficiently [1,2,3,4,5]. Using classical computers, the computational cost of performing electronic structure calculations that rely on solving the Schrodinger equation grows exponentially with respect to system size. Alternatively, quantum computing can potentially address this exponential cost problem by utilizing the collective properties of quantum states, including superposition, interference, and entanglement, to store the wave function using a linear number of qubits. In a seminal work, Peruzzo et al. reported the use of iterative quantum phase estimation (QPE) algorithms to handle electronic correlation effects given a numerically best wave function within the space spanned by the basis set [6]. While an advantage in computational scaling against its classical counterpart is achieved, the long circuit depths required to generate the wave function causes QPE’s accuracy to be severely constrained by the physical characteristics of currently available noisy intermediate-scale quantum (NISQ) devices such as coherence time issues and error rates. Though the mitigation of these physical limitations via the implementation of a fault-tolerant scheme has been reported, the resource requirements are still too large for practical deployment [7,8,9]. 

In contrast, the use of the hybrid quantum–classical variational quantum eigensolver (VQE) algorithm to solve the electronic structure problem is considered the most suitable model for chemical application, as this algorithm mitigates the significant hardware demands needed by QPE on NISQ devices [10,11]. VQE is based on the Rayleigh–Ritz variational principle:(1)$$E\leq \langle \Psi │\hat{H}│\Psi \rangle$$
where Ψ is the molecular wave function, $\hat{H}$ is the electronic Hamiltonian, and E is the ground state energy. The electronic Hamiltonian is written in the second quantized form:(2)$$\hat{H}=\sum_{pq}h_{pq}{\hat{a}}_{p}^{†}{\hat{a}}_{q}+\frac{1}{2}\;\sum_{pqrs}g_{pqrs}{\hat{a}}_{p}^{†}{\hat{a}}_{q}^{†}{\hat{a}}_{r}{\hat{a}}_{s}$$
where $h_{pq}$ and $g_{pqrs}$ represent one- and two-body integrals and ${\hat{a}}_{p}^{†}$ and ${\hat{a}}_{p}$ are anti-commuting creation and annihilation operators, respectively. Equation (2) is mapped to a qubit Hamiltonian by using one of three encoding methods: Jordan–Wigner, parity, or Bravyi–Kitaev [12,13,14]. VQE uses a quantum processing unit (QPU) to prepare a parametric version of either a heuristic ansatz that is easily implemented on a real quantum device or a chemistry-inspired trail ansatz, $\Psi (\vec{\Theta })$. This trial ansatz depends on a vector of real-valued variational parameters, $\overset{⃑}{\Theta }=\{\theta_{i}\}$, which is variationally tuned to minimize the energy expectation value. On the other hand, a classical processing unit is used to collect quantum computer data and optimize the parameters within the variational loop. The number of qubits required for the quantum calculation is dependent on the number of spin orbitals considered within the chemical system in question. A pioneering application of VQE is its examination of the ground-state properties of H_2_ using 2 qubits [6]. The method has also been extended to other molecular applications (LiH, H_2_O, BeH_2_, NH_3_, CH_4_, and CO), with a simulation scale ranging from 4 to 20 qubits [15,16,17,18]. In these studies, calculated ground-state energies achieve chemical accuracy when compared to results computed using a classical matrix eigenvalue decomposition and full-configuration interaction (FCI) method. The simulation of an even larger C_2_H_4_ molecule requiring 28 qubits with reasonable computing resources has also been demonstrated by employing group symmetry to significantly reduce the circuit depth of the ansatz [19].

Despite progress for simple molecular systems, application of VQE to describe realistic chemical systems with technological applications remains challenging in the foreseeable future, as larger systems demand an impractical number of qubits, rendering the calculations prohibitive and inferior to classical methods. Although progress has been made in scaling up hardware with respect to the number of available qubits, short coherence time and sensitivity to the external environment are still limiting factors for performing quantum computing calculations on large chemical systems. The concomitant drastic accumulation of noise requires expensive mitigation protocols to reach the desired level of accuracy [8,20]. To make VQE practical in the near term, hardware-friendly multi-scale hybrid quantum–classical embedding algorithms have been proposed to partition large-scale systems into fragments, confining the application of quantum simulation to more manageable active regions [16,21,22,23,24,25,26]. A particular category of embedding consists of reducing the molecular orbitals to a subset of active orbitals that provide a reliable description of the static electronic correlation, and this quantum subsystem is then embedded in the mean field generated by the inactive electrons [27]. Within this scheme, the expectation value of the Schrodinger equation in the active space is evaluated over a quantum circuit using VQE on a quantum simulator or quantum device. The rest of the system is then treated with classical Hartree–Fock (HF) [28] or density functional theory (DFT) methods. Such an embedding approach has been employed previously to investigate the ground-state properties of H_2_O and the dissociation profiles of N_2_ and O_2_ [27] (See Appendix A). Additionally, within this study the energetics associated with the cleavage of the C–C bond in C_2_H_4_O was examined; a reaction system beyond the reach of all-electron VQE in the absence of hardware which provides the resources required to handle the large number of qubits and circuit depth representation. This technique, in combination with algorithms to reduce circuit depth, is also used to simulate the triple bond-breaking process in butyronitrile [29]. From a chemistry perspective, technologically relevant reactions entail more than the simple stretching of covalent bonds or changing of bond angles. Recently, we utilized this embedding approach to examine the coupling of CO_2_ and NH_3_, a representative complex reaction that leads to the formation of the NH_2_-COOH species [30,31]. When referenced to results calculated using classical coupled cluster with single and double substitutions (CCSD), the embedding approach performs better than HF in calculating a reaction profile despite the drastic reduction in qubit resources.

Further application of this embedding algorithm to study complex reactions has received limited attention. This study aims to extend the embedding algorithm within the VQE framework to hydride complexes of Li. We also previously reported the first-of-its-kind all-electron VQE simulations of the ground-state properties of LiH_n_ (n = 1–3) species, including their singly charged ions [31]. The present manuscript is an extension of our prior studies, describing the application of the embedding approach in the simulation of the hydrogenation of Li_n_ (n = 2–4) clusters. This reaction system provides a simplified model for exploring the interaction between hydrogen atoms and metals, and can be applied to the chemisorption of hydrogen on metal surfaces [32]. This study could also serve as a baseline testbed in understanding previously reported results suggesting the potential of using Li-based metal systems as hydrogen storage media [33,34,35]. An additional aim is to further benchmark the embedding approach with respect to classical multireference ab-initio methods for reactions requiring an accurate description of bond breaking and formation.

## 2. Methodology

In preparation for the embedding calculations, geometries of all local minima (reactants and products) associated with the Li_2_ + H_2_, Li_3_ + H_2_, and Li_4_ + H_2_ reactions were optimized using classical spin-polarized plane-wave DFT, as implemented in the Vienna Ab-Initio Simulation Package (VASP) version 5.4.4 [36,37]. Perdew–Burke–Ernzerhof (PBE) [38] functional plane-wave basis sets with a cutoff energy of 520 eV and projector augmented wave [39] pseudopotentials [40] were employed. The self-consistent simulated valence electrons were taken to be 1*s*^1^ for H and 2*s*^1^ for Li. A three-dimensional 30 Å × 30 Å × 30 Å periodic box was inserted in each simulation model to exclude artificial periodic interaction. The sampling of the Brillouin zone was conducted with a Γ-point *k*-point mesh. The ionic and electronic convergence limit was set to 0.03 eV/Å and 1 × 10^−5^ eV, respectively. The Methfessel–Paxton scheme [41] was utilized with a modest smearing width of 0.1 eV, and the total energies are extrapolated to σ → 0. 

To determine the reaction path and to locate the transition-state structures along the H_2_ dissociation routes, the climbing-image nudged elastic band method (CI-NEB) was employed [42]. Thirteen-to-nineteen images interpolated between the initial and final states were used. The reaction barriers were referenced to initial states constructed by placing a molecular H_2_ on the clusters which was then fully relaxed using DFT-PBE to obtain weakly adsorbed H_2_ models. The geometries of the final states are based on previously reported most stable structures of Li_2_, Li_3_, Li_4_, Li_2_H_2_, Li_3_H_2_, and Li_4_H_2_, where the adsorbed H atoms in the hydrogenated species are two-fold coordinated to two different cluster sites [32,43,44]. Geometry re-optimization using plane-wave DFT-PBE was conducted prior to the CI-NEB calculations. Use of plane-wave DFT in combination with CI-NEB for reaction pathway preprocessing has been adopted in previous studies [45].

At each of the DFT-PBE-optimized structures along the reaction pathway, single-point quantum computing calculations were performed, and the results were compared to those obtained from classical HF and post-HF quantum chemistry methods. Here, we used a quantum embedding scheme which allows for the treatment of a select number of electrons and orbitals to facilitate the simulation. In this framework, the electronic structure of the full system is broken into fragments consisting of (i) the active orbitals, which define a subset of valence electrons and frontier orbitals, and (ii) the environment [27]. Each region is described quantum mechanically with the environment treated using classical HF, while employing a post-HF quantum mechanical description of the active orbitals which is carried out on the quantum simulator. The time-independent Schrödinger equation of the active space is connected to the exchange-correlation embedding potential of the environment, such that a new Hamiltonian is defined for the full system. A reduction in qubit resources is achieved since the VQE computation is restricted to the active orbitals.

The embedding calculations were carried out using the Qiskit Nature platform, a Python package that interfaces the quantum computing framework with the existing classical quantum chemistry software PySCF v2.6.2 to generate classical data such as electronic integrals in the atomic orbital basis [46,47]. The molecular orbitals were prepared by performing restricted open-shell HF calculations on the optimized geometries, and an active space with *m* electrons, *n* occupied orbitals, and *p* virtual orbitals (AS(*m*,(*n + p*)) was selected. The active space is identified by first looking at the active electrons involved in the reaction. The orbitals involved two σ bonds, one each from H_2_ and the considered Li clusters. As seen below, these undergo conversion into essentially two partial σ bonds between the two species in the transition state, prior to conversion into complete bonds in the product. Overall, this process involves at least a 4-electron and 4-orbital active space, AS(4e, 4o). Orbitals beyond this baseline are selected next, as the accuracy of quantum embedding generally improves with an increasingly larger active space [27]. We managed to push our hardware to use active spaces that require up to 14 qubits in a reasonable amount of time and memory. A unitary coupled cluster Ansatz with single and double excitations (UCCSD) was used to represent the electronic wavefunction in the active space and the Jordan–Wigner scheme is used to map the wavefunction onto qubits [3]. VQE simulations were carried out on the Qiskit statevector simulator along with the STO-6G basis set [28]. The gradient-based Broyden–Flecher–Goldfard–Shanno minimization algorithm (BFGS) was used for the energy minimization for calculations on the simulator [48]. 

## 3. Results and Discussion

The structures of all reactant and product clusters considered in generating reaction profiles are displayed in Figure 1. These geometries are taken from the lowest energy-stable configurations identified from previous exhaustive studies using classical DFT techniques [32,43,44]. For Li_2_, only one trivial linear configuration is possible. For Li_3_ and Li_4_, planar structures with C_2v_ and D_2h_ symmetries are reported, respectively. The geometry of the lowest energy structure of Li_2_H_2_ is a rhombus, while that for Li_3_H_2_ can be viewed as a deformed trapezoid. For Li_4_H_2_, the lowest energy predicted ground-state configuration is planar, two-dimensional and consisting of a distorted rhombus Li_4_ with the hydrogen atoms bonded on two adjacent sides of one Li atom site. 

Spot checks were conducted to initially validate the embedding approach. In particular, the ground-state energies of Li_4_ and the selected low-lying energy isomers of Li_4_H_2_ were calculated (see Figure 1). Predicted energy variations are compared using HF and post-HF methods such as CCSD, complete active space configuration interaction (CASCI), and complete active space self-consistent field (CASSCF), as implemented in PySCF. Table 1 summarizes the results calculated using the STO-6G basis set. In agreement with post-HF methods, the embedding approach predicts that the modified version of Li_4_ (Li_4_(I); see Figure 1) is slightly less stable. For all Li_4_H_2_ isomers considered, the embedding method used here yields the correct stability trend: Li_4_H_2_ > Li_4_H_2_(I) > Li_4_H_2_(II). The Li_4_H_2_(I) isomer, whose metal atoms do not lie in the same plane, is a critical case since it exhibits quasi-degeneracy within less than 3 milli-Hartree (mHa) from CCSD calculations. The embedding approach can unambiguously predict such property. Additionally, it outperforms the conventional HF for all considered isomers, despite the drastic reduction in the number of qubits.The predicted energies are within a few 10 mHa from the refence CCSD, and the discrepancy is much lower when compared to CASCI and CASSCF. In particular, the energy difference is not more than 1 mHa with respect to CASCI.

With geometries from classical computation in hand, we then turned to calculations using a quantum simulator to determine energies and reaction profiles. The potential energy curve for the Li_2_ + H_2_ reaction calculated from the embedding approach is shown in Figure 2. The relative energy is defined as *E*_INIT_ − *E*_TS_, where the initial and transition state energies are calculated using the different methods, respectively. In the initial state, the bond axis of the hydrogen molecule is nearly perpendicular to Li–Li, with the dissociation initiated via the H_2_ migration toward the cluster. At the transition state, the H–H bond distance is lengthened by ~0.2 Å relative to the initial state, indicating that the molecule starts dissociating over Li_2_. The process does not pass through a further stationary point, and, thus, the conversion to the Li_2_H_2_ product occurs in a single concerted step. 

Figure 2 shows comparisons of the relative energies computed using classical muti-reference full-space FCI, as well as CASCI and CASSCF in which similar active spaces are used as that in the embedding approach. All methods predict a kinetically activated and thermodynamically downhill process. The embedding method using an active space of 14 qubits (AS(4e, 7o)) is in good agreement with the FCI curve, albeit not within the desired chemical accuracy of 1 mHa. As the dissociating H_2_ moves toward Li_2_, the embedding curve is within 4 mHa of FCI. A larger deviation is observed at the transition state where strong correlation effects are significant, with embedding slightly overestimating the barrier by 33 mHa. As the reaction proceeds to product, the embedding curve remains mostly parallel to FCI with an energy difference in the 10–20 mHa range. With respect to CASCI, the embedding method displays relatively lower discrepancy in energy across the entire reaction profile. The predicted barrier overestimates the CASCI value by a smaller value of 26 mHa, while the energy deviation outside the transition state is not more than 2 mHa. We note that the CASSCF curve differs considerably from embedding, FCI, and CASCI. In particular, the method yields a visible discontinuity in the pre-transition state region of the curve, where H_2_ begins to dissociate, and each H atom begins to form a bond with Li_2_. Unlike CASCI, CASSCF orbitals tend to localize due to additional orbital coefficients optimization, leading to electrons preferentially correlated in a region of space over another [49,50]. While it results in the further lowering of the total energy, the unphysical localization causes the wavefunction to change discontinuously along the section of the reaction path, where bond breaking and new bond formation begin. The resulting discontinuity in the energy is quite discernable due to the larger nearly degenerate virtual orbitals present in the active space. One can get around this issue by rotating the necessary orbitals with virtuals outside of the used active space, however the same active space was used for the embedding, CASCI and CASSCF calculations throughout this study. 

The dissociation on Li_3_ starts with H_2_ physiosorbed on a single Li site (Figure 3). The H–H bond length then expands as the molecule approaches the substrate. At the transition state, H_2_ is essentially cleaved, and both H atoms stay near the initial Li site, concurrently forming H–Li bonds with a bond length of 1.70 Å. Finally, both H atoms move away from each other and toward the adjacent bridge Li atoms. Embedding using an active space of (AS(5e, 7o)) predicts an exothermic process with a reaction energy of −43 mHa. The activation barrier of 99 mHa corresponds to the H–H bond cleavage and formation of Li–H bonds with subsequent steps not transiting through a further stationary point. By comparison, FCI exhibits a transition state occurring a step earlier, with a lower barrier than the embedding approach. We note that the structural difference in the transition states is minimal in that the H_2_ molecule is cleaved, and the variation in the Li–H bond length is <0.03 Å. Moreover, both embedding and FCI methods show a qualitative consistency across the potential energy curve. Both embedding and CASCI methods yield a similar location of the transition state and calculated energy barrier, overestimating the FCI barrier by 20 mHa. The transition state location predicted by CASSCF is more consistent with FCI, but it underestimates the FCI barrier by 20 mHa. The overall variation between the two curves is also more discernable. As in the Li_2_ + H_2_ case, the level of agreement between embedding and CASCI is relatively better.

The potential energy curve for H_2_ dissociation on Li_4_ with an irregular rhombic molecular geometry is shown in Figure 4a. Initially, H_2_ is weakly physiosorbed at a single Li site. Bond breaking between the H–H bond then occurs as the molecule approaches the Li_4_ cluster. Simultaneously, Li_4_ folds along its short diagonal to facilitate the addition of H atoms. At the transition state, the two triangles sharing a common edge in Li_4_ are oriented perpendicular to each other, while the H atoms bind to adjacent bridge sites. Finally, Li_4_H_2_ reorients toward the most stable rhombic configuration. Here, embedding calculates an activation barrier of 62 mHa and a reaction energy of −58 mHa. Alternatively, H_2_ can also be weakly adsorbed along the short diagonal of Li_4_, parallel to the cluster plane (Figure 4b). Proceeding from the initial step, H_2_ gradually moves toward Li_4_, and the molecule interacts with adjacent Li atoms at the transition state. It then begins dissociating as it moves away from the short diagonal, forming the rhombic Li_4_H_2_ product. This path is also exothermic but leads to a barrier of 46 mHa, which is less kinetically activated than starting from a weakly H_2_ physiosorbed at a single Li site (Figure 4a).

The embedding scheme predicts potential energy surfaces with double wells due to the presence of an additional product channel. For the first pathway (Figure 4a), this additional channel involves a H–Li bond formation as H_2_ approaches the cluster to produce a preliminary H_2_–Li_4_ pair. Embedding finds the process to be slightly downhill (−2 mHa), with a barrier of 31 mHa. This intermediate then undergoes an isomerization step in which the scission of the adsorbed H–H bond is followed by H atom migration to the adjacent Li–Li sides to yield the final product. For the second pathway (Figure 4b), the additional product channel consists of each H atom three-fold bonded to the cluster. The barrier height associated with the subsequent isomerization step is 9 mHa, which is 37 mHa smaller than the transition state for the preceding step. 

The potential energy curves obtained using the classical CCSD method are provided in Figure 4a,b. Due to the large computational cost, FCI can only be calculated for the relatively smaller H_2_ + Li_2_ and H_2_ + Li_3_ reaction systems. The embedding generally compares well with the CCSD curves. The prediction that hydrogenation of Li_4_ transits through an additional stationary point is consistent with CCSD, although the barriers are slightly overestimated. Both embedding and CCSD predict that the hydrogenation reactions considered are thermodynamically downhill and that the second pathway (Figure 4b) is more kinetically favorable. For the H_2_ + Li_2_ and H_2_ + Li_3_ reaction systems, we find that embedding curves follow CASCI over the entire region and that the more expensive CASSCF treatment yields an inferior quantitative agreement. These trends are also observed for the hydrogenation of Li_4_.

The total numbers of qubits and excitation parameters (*N_ex_*) utilized by the embedding method for the various reaction systems in comparison to conventional VQE were calculated (see Table 2). Assuming a singlet state for Li_2_ + H_2_ and Li_4_ + H_2,_
*N_ex_* is calculated as the sum of single- and double-excitation terms: *N_ex_* = *n_occ_n_vir_* + *n_occ_n_vir_* (*n_occ_n_vir_* + 1)/2, where *n_occ_* and *n_vir_* represent the number of occupied and unoccupied spatial orbitals [51]. For Li_3_ + H_2_, the corresponding excitation parameters are multiplied by 2, assuming a doublet state (½ or −½). Table 2 shows a significant reduction in computational resources when embedding is used. Compared to conventional VQE, embedding uses about 40–70% less qubits, while the reduction is up to several orders of magnitude for the excitation parameters. 

The calculations presented here are performed on a quantum simulator, and calculations on quantum computers would need to be performed to further assess the viability of the embedding algorithm for modeling complex chemical reactions. The current hardware requires the use of error mitigation techniques, and as these techniques mature, the desired simulation accuracy could be achieved [52,53]. A viability assessment on quantum computers is beyond the scope of this work and will become the subject of future investigations.

## 4. Conclusions

While quantum computing is considered a paradigm shift in our basic understanding of physical computation, effective implementation of quantum computing in practical applications also depends on progress and development in the dimensions of both quantum computing hardware and quantum computing algorithms. From the perspective of quantum computing hardware, the availability of the number of qubits and the noise level of the qubits should be weighed, whereas from the perspective of quantum computing algorithms, error tolerance capability in the algorithm and gain of speed-up relative to classical computing should be considered. In addition, current quantum processing devices and quantum computing algorithms may also require pre- and post-processing using classical computers for basic operation within realistic architecture. 

Although quantum computers have already been used to model chemical reactions, and as this technology continues to develop, it will have transformative implications for material design and discovery. The ability to model large systems and rapidly screen material properties will significantly benefit many application fields. Due to the limitation of the quantum device and available qubits, at current stage, the classical–quantum hybrid approach is a practical way to solve real problems. To build such hybrid solutions, in this work, we presented some insights into the quantum active space-embedding approach, which seeks a balance between accuracy and computational cost. We particularly focused on its deployment to the calculations of the activation and reaction energies of the hydrogenation of Li_2_, Li_3_, and Li_4_ clusters as a testbed for performance evaluation. These are prototypical examples of a complex chemical reaction between two reactants reacting in synchronous fashion. The significant structural reorganization that occurs during bond breaking and formation is a compelling testbed for validation, offering data for future refinement. The considered reactions involve a transition state which is typically more sensitive to approximations in the solution of the Schrodinger equation than reactants and products, due to the presence of partial bonds. The quantum calculations, using the statevector simulator, successfully map out each process over the entire potential energy curve. The predicted potential energy curves qualitatively reproduce the classical FCI results for H_2_ + Li_2_ and H_2_ + Li_3_ reactions in both the bond-breaking and bond-formation regions. A similar trend is found for the corresponding hydrogenation of Li_4_ when comparing the curves acquired using the embedding and CCSD approach. The error notably increases as the reaction proceeds to the transition state, which could be attributed to its far complex electronic structure in contrast to reactants and products. The embedding results show better accuracies when compared to the CASCI method, which deploys the same active orbitals. Similarly, the embedding approach shows qualitative advantages over CASSCF in the description of the reaction profiles of the hydrogenation of the considered Li clusters, though this is attributed to the active space used, as mentioned previosuly. Our results confirm the qualitative viability of the deployed embedding approach for mapping out reaction profiles for molecular systems, while providing data to compare against future calculations, using different flavors of embedding schemes and positioning the approach as a focal point for further ongoing development. Our study also indicates that it is possible to perform quantum computing on large reaction systems for practical applications.

## Figures and Tables

**Figure 1 nanomaterials-14-01267-f001:**
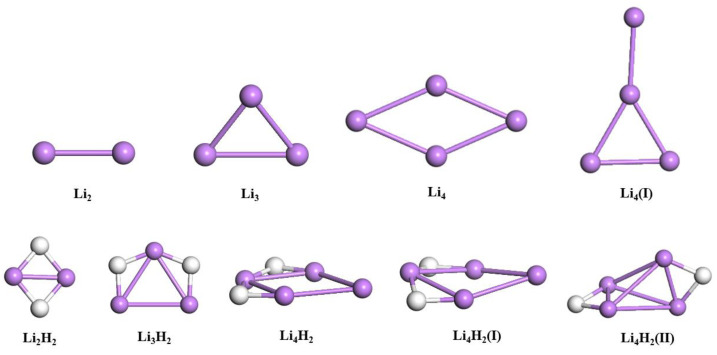
Geometries of reactant and product clusters. Li_4_(I) and Li_4_H_2_(I-II) are low-lying isomers. Atom colors: Li, purple; and H, white.

**Figure 2 nanomaterials-14-01267-f002:**
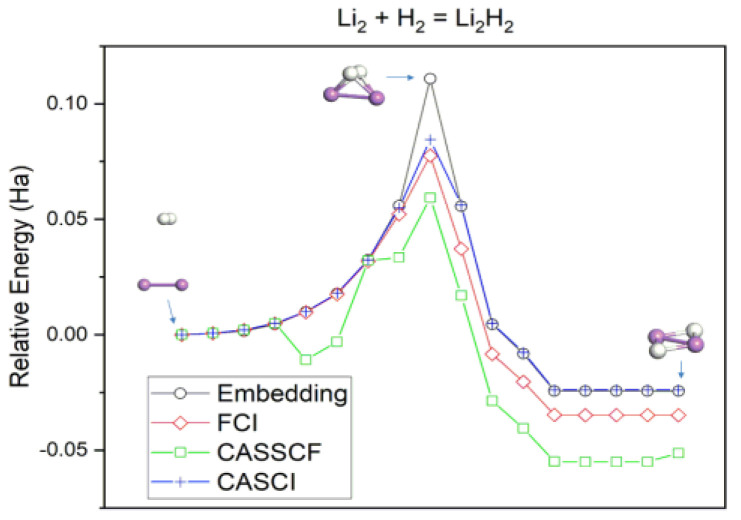
Potential energy curves for the hydrogenation reaction of Li_2_ with H_2_. The atom coloring scheme follows the one in Figure 1. CASCI and CASSCF calculations utilize the same active orbitals in the embedding approach (AS(4e,7o)).

**Figure 3 nanomaterials-14-01267-f003:**
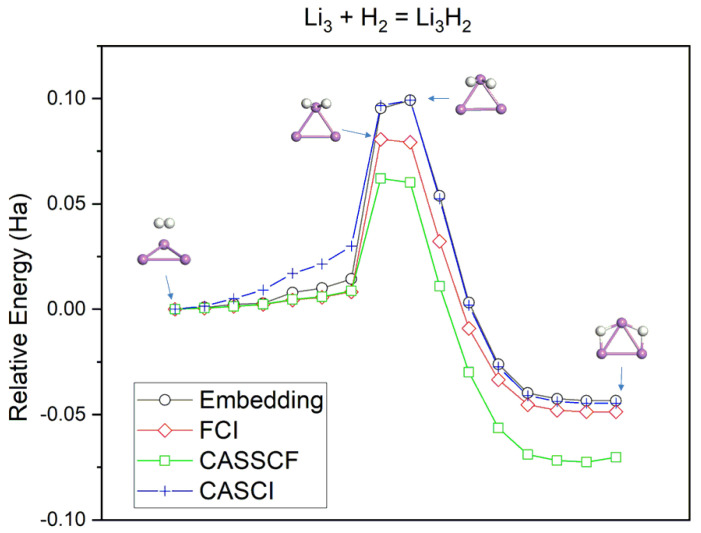
Potential energy curves for the hydrogenation reaction of Li_3_ with H_2_. The atom coloring scheme follows the one in Figure 1. CASCI and CASSCF calculations utilize the same active orbitals in the embedding approach (AS(5e,7o)).

**Figure 4 nanomaterials-14-01267-f004:**
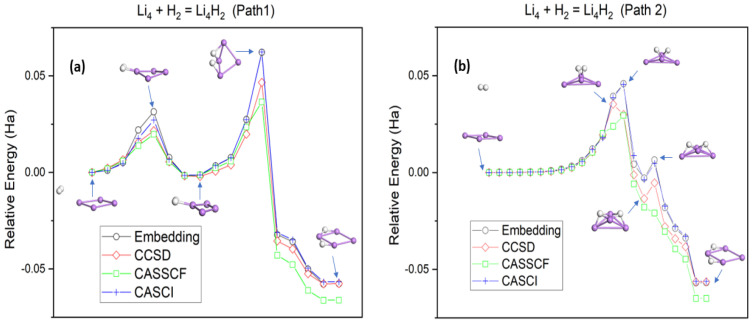
Potential energy curves for the hydrogenation reaction of Li_4_ with H_2_. (**a**) Path1: H_2_ is initially physiosorbed at a Li site. (**b**) Path2: H_2_ is initially above the Li_4_ plane. The atom coloring scheme follows the one in Figure 1. CASCI and CASSCF calculations utilize the same active orbitals in the embedding approach (AS(6e,7o)).

**Table 1 nanomaterials-14-01267-t001:** Quantum embedding, HF, CASCI, CASSCF, and CCSD energies in Ha computed with STO-6G basis set. CASCI and CASSCF calculations utilize the same active orbitals in the embedding approach (AS(4e,6o) and AS(6e,7o) for Li_4_ and Li_4_H_2_).

	Embedding	HF	CASCI	CASSCF	CCSD
Li_4_	−29.6701	−29.6482	−29.6705	−29.6927	−29.7072
Li_4_ (I)	−29.6444	−29.6247	−29.6447	−29.6653	−29.6834
Li_4_H_2_	−30.8487	−30.8313	−30.8494	−30.8741	−30.9033
Li_4_H_2_ (I)	−30.8460	−30.8285	−30.8466	−30.8712	−30.9006
Li_4_H_2_ (II)	−30.8062	−30.7933	−30.8072	−30.8550	−30.8690

**Table 2 nanomaterials-14-01267-t002:** Qubits and excitation parameters (N_ex_) used by the quantum embedding method vis-à-vis conventional VQE for the reaction systems considered.

System	Electrons	Molecular Orbitals	Method	Qubits	Parameters
Li_2_ + H_2_	8	12	VQE	24	560
			Embedding	14	65
Li_3_ + H_2_	11	17	VQE	34	4554
			Embedding	14	180
Li_4_ + H_2_	14	22	VQE	44	5670
			Embedding	14	90

## Data Availability

The data that support the findings of this study are available within the article. Structural geometries involved in the various potential energy curves are available upon request.

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
