# Peer review of "How Well Can Quantum Embedding Method Predict the Reaction Profiles for Hydrogenation of Small Li Clusters?"

_nanomaterials, 2024, doi:10.3390/nano14151267_

Round 1
Reviewer 1 Report
Comments and Suggestions for Authors
This is an interesting article dealing with quantum computing on hydrogenation of small Li n clusters (n=1,4). It is basically a benchmark test for the quantum computing in predicting chemical reaction. Authors reported energetic values for comparison with results from conventional quantum chemical schemes. As Authors stated that the values obtained in their calculations are still out of chemical accuracy. But Authors did not report any values concerning computational cost. It is not easy to evaluate its credit.
Author Response
Comment: This is an interesting article dealing with quantum computing on hydrogenation of small Li n clusters (n=1,4). It is basically a benchmark test for the quantum computing in predicting chemical reaction. Authors reported energetic values for comparison with results from conventional quantum chemical schemes. As Authors stated that the values obtained in their calculations are still out of chemical accuracy. But Authors did not report any values concerning computational cost. It is not easy to evaluate its credit.
Response: Following Li et al. (Chem. Sci. 13, 8953 (2022)), the total number of qubits and excitation parameters (Nex) utilized by the embedding method for the various reaction systems in comparison to conventional VQE are calculated. Assuming a singlet state for Li2 + H2 and Li4 + H2, Nex is calculated as the sum of single- and double-excitation terms: Nex = noccnvir + noccnvir(noccnvir + 1)/2, where nocc and nvir represent the number of occupied and unoccupied spatial-orbital. For Li3 + H2, the corresponding excitation parameters is 2 Nex assuming a doublet state (½ or -½). Results show a significant reduction in computational resources when embedding is used. Compared to conventional VQE, embedding uses about 40 - 70% less qubits while the reduction is up to several orders of magnitude for the excitation parameters.
Changes in the paper: The above findings and the numerical results summarized in a new Table 2 are included on page 8 of the revised manuscript.
Reviewer 2 Report
Comments and Suggestions for Authors
This is a fairly straightforward proof-of-concept paper applying the VQE method to the hydrogenation of small Li clusters, which presents another example for a VQE quantum simulation. On those grounds it is novel but does not really add anything new. A comparable paper is that of Rossmannek et al. (https://pubs.acs.org/doi/10.1021/acs.jpclett.3c00330) which really should be cited as not only are they using their modified version of the same method albeit to a different problem, they have put the simulation to the test on a real quantum computer "The Qiskit IBM Runtime service" which shows that the "viability" posed in this paper may not translate to reality - the noise problem.
From the perspective external to presenting a toy problem for quantum computing simulations I found this paper a little odd, especially from my background in classical quantum chemistry. No-one in my field would use the approach taken by the authors to study the problem tackled.
I was perplexed by the extensive description of the VASP method to obtain geometries on which they then used a wavefunction based method. I am not as precious as some referees in my field who insist geometries should be optimised at the same level as the subsequent method used for properties but I feel that planewave vs wavefunction is too different. The authors did not provide geometries - which they should have in supplementary material for me to convince myself that they are appropriate. This was especially because all the Figures made me wonder about the appropriateness of the active space. I would have expected some description of choice of the CAS space and in the end had to run my own calculations to see whether AS(4e,6o) and AS(6e,7o) were sensible. They seemed to be but wouldn't be for the wrong geometry hence increasing my curiosity about the geometry.
There was mention of FCI in the text but not the table so I found that a bit puzzlingm too.
So as a paper for reaction profiles of Lithium clusters I don't think this will add to the canon. As another example of how quantum computers might work it is probably acceptable but not really bringing anything new to the table. I think that would need the results from running on a real quantum computer.
Author Response
Comment: This is a fairly straightforward proof-of-concept paper applying the VQE method to the hydrogenation of small Li clusters, which presents another example for a VQE quantum simulation. On those grounds it is novel but does not really add anything new. A comparable paper is that of Rossmannek et al. (https://pubs.acs.org/doi/10.1021/acs.jpclett.3c00330) which really should be cited as not only are they using their modified version of the same method albeit to a different problem, they have put the simulation to the test on a real quantum computer "The Qiskit IBM Runtime service" which shows that the "viability" posed in this paper may not translate to reality - the noise problem.
Response: The application portions of Rossmannek’s previous two papers focus on looking at the dissociation profiles of test molecular systems. As mentioned on pages 2-3 of the manuscript, our work goes beyond such simple simulation by exploring the use of quantum embedding to model chemical reactions as it is an anticipated key application of quantum computing. In particular, we selected a prototypical example of a complex chemical reaction, between two reactants reacting in synchronous fashion. The significant structural reorganization that occurs during bond breaking and formation is a compelling testbed for validation, offering data for future refinement. The considered reactions involve a transition state which is typically more sensitive to approximations in the solution of the Schrodinger equation than reactants and products, due to the presence of partial bonds. The paper showcases the potential of the current embedding algorithm to ascertain important trends, positioning the approach as a focal point for further ongoing development.
The referee is correct to point out that the noise problem could compromise viability. A parallel activity in quantum computing is the development of error mitigation techniques, and as these techniques become mature, desired simulation accuracy could be achieved (see e.g., Suzuki et al, PRX Quantum 3, 013405 (2022); Van den Berg et al., Nature Physics, 19 1116 (2023)).
Changes in the paper: Following the author’s recommendation, we cited Rossmanek’s work mentioned above and also described it briefly on page 2. We further firmed up the contribution of our work in the area in the Conclusion section on page 9. The error mitigation techniques are also mentioned on page 8.
Comment: From the perspective external to presenting a toy problem for quantum computing simulations I found this paper a little odd, especially from my background in classical quantum chemistry. No-one in my field would use the approach taken by the authors to study the problem tackled. I was perplexed by the extensive description of the VASP method to obtain geometries on which they then used a wavefunction based method. I am not as precious as some referees in my field who insist geometries should be optimised at the same level as the subsequent method used for properties but I feel that planewave vs wavefunction is too different.
Response: We have found a recent paper which essentially use protocol that we have implemented in this work (Christopoulou, et al, Phys. Chem. Chem. Phys. 26, 5895-5906 (2024)). As in our work, the overall strategy involves initial preprocessing with a classical plane-wave DFT code in combination with Nudged Elastic band (NEB) technique to generate the potential energy surface for the reaction, and then using the structural data for performing computations with quantum algorithms. The same structural data is also used to perform reference post-HF calculations. The NEB algorithm available in most plane-wave DFT codes is widely used in computational catalysis community due its effectiveness in finding the minimum energy paths between known reactants and products. To ensure energy convergence, a large energy cutoff is imposed on the wavefunction.
Changes in the paper: The revised version of the paper mentions the above paper in the Methodology section on page 3.
Comment: The authors did not provide geometries - which they should have in supplementary material for me to convince myself that they are appropriate. This was especially because all the Figures made me wonder about the appropriateness of the active space. I would have expected some description of choice of the CAS space and in the end had to run my own calculations to see whether AS(4e,6o) and AS(6e,7o) were sensible. They seemed to be but wouldn't be for the wrong geometry hence increasing my curiosity about the geometry.
Response: The active space is identified by first looking at the active electrons involved in the reaction. The orbitals involved two r bonds, one each from H2 and the considered Li clusters. As seen in the paper, these undergo conversion into essentially two partial r bonds between the two species in the transition state, prior to conversion into complete ones in the product. Overall, this process involves at least a 4 electron and 4 orbital active space, AS(4e, 4o). We next selected ones beyond this baseline as the accuracy of quantum embedding generally improves with increasingly larger active space (Rossmanek et al., J. Chem. Phys. 154, 114105 (2021). We managed to push our hardware to use active spaces that require up to 14 qubits in a reasonable amount of time and memory.
Changes in the paper: We included the above description on page 4 of the revised manuscript. Following the recommendation of the referee, a separate Supplementary Information material is also added containing the atomic coordinates relevant to energy profiles of the reactions considered here.
Comment: here was mention of FCI in the text but not the table so I found that a bit puzzling too.
Response: The molecular systems enumerated in Table 1 were not subjected to FCI calculations as the resource requirements are too large for our local machines. Instead, the alternative CCSD method was used.
Comment: So as a paper for reaction profiles of Lithium clusters I don't think this will add to the canon. As another example of how quantum computers might work it is probably acceptable but not really bringing anything new to the table. I think that would need the results from running on a real quantum computer.
Response: Though our calculations are done on a noiseless quantum simulator, our results reinforce the potential utility of partitioning the fermionic Hamiltonian into an active orbital (treated by quantum algorithm) and environment portion (solved classically) to treat problems that are beyond the reach of conventional VQE. As mentioned in the Conclusion section, of particular relevance are the results obtained for the hydrogenation of the considered Li clusters which demonstrate the applicability of the deployed quantum embedding scheme to non-trivial chemical reactions. Our work also provides data for comparison with any future iteration of this technique.
We agree with the referee that calculations on quantum computers would need to be done for further viability assessment. However, this goes beyond the scope of this work and will become the subject of future investigations.
Changes in the paper: One page 8 of the revised manuscript, we mentioned the need to perform the calculations on quantum computer to further evaluate the viability of the deployed quantum embedding algorithm.